# Tell Me What You Waste and I'll Tell You Who You Are: An Eight-Country Comparison of Consumers' Food Waste Habits

Elisa Iori [1], Matteo Masotti [1,*], Luca Falasconi [1], Enzo Risso [2], Andrea Segrè [1] and Matteo Vittuari [1]

1 Department of Agricultural and Food Sciences, University of Bologna, 40127 Bologna, Italy
2 IPSOS s.r.l., via Tolmezzo 32, 20132 Milano, Italy
* Correspondence: matteo.masotti8@unibo.it

**Abstract:** Using an original survey conducted in eight countries in 2021 (Canada, China, Germany, Italy, Russia, Spain, the UK, and the USA), this study explored the relationship between household food waste and dietary habits through a cross-country comparative perspective. In total, 8000 questionnaires were recorded from samples representative of the adult population of each country through an online survey conducted between the 13th and the 24th of August. The questionnaires were developed from the Waste Watcher International Observatory on Food and Sustainability, an international study of the social, behavioral, and lifestyle dynamics behind household food waste. The relationships between the per capita self-reported amount of food waste (expressed in kilocalories) and self-declared dietary habits (traditional, healthy and sustainable, vegetarian, smart, and confused) were estimated using multiple linear regression models. The results showed that smart diets are associated with higher values of food waste in Canada, Spain, the UK, and the USA. Vegetarian diets are associated with lower food waste values in China, Germany, the UK, and the USA, but not in Italy, Russia, and Spain. The share of the population adopting a smart diet was, on average, 2.7% of the sample; therefore, interventions for food waste reduction should focus on these specific types of consumers, who are often associated with larger amounts of food waste.

**Keywords:** household food waste; diets; food choices; dietary patterns; sustainable behaviors; sustainable consumption

## 1. Introduction

Food waste is recognized as one of the most important global manifestations of the inefficiencies of food systems. The UN Food System Summit 2021 emphasized the multiple impacts of food waste in terms of environmental and economic impacts. Its call for action has been further spotlighted by the UN Framework Convention on Climate Change's 26th Conference of the Parties (COP26), which emphasized how climate change responses require the coupling of public interventions with individual actions. Additionally, EUROSTAT estimated that in 2020, approximately 127 kg of food waste was generated per inhabitant in the EU, making a total of almost 57 million tons across all the member states [1].

A large part of the literature is dedicated to consumers being responsible for producing a higher proportion of food waste both in industrialized [2] and non-industrialized countries [3]. As a result, growing attention has been paid to the consumption stage, which has been recognized as an essential behavioral issue where multiple inter-related and competing drivers play an influential role [4].

Food waste behavioral drivers include personal preferences such as taste and cultural issues [5]; habits such as the frequency of shopping [6]; attitudes related to thoughts and feelings [7]; social norms [8]; and knowledge and skills. Food can be wasted due to poor understanding of date labels or of food storing practices aiming to maximize shelf life [9].

Similarly, food waste can be related to a lack of ability in food preparation and portioning [10,11]. Some attention has been given to consumers' individual concerns regarding food waste implications, and therefore, to their understanding of the environmental [12], economic, and social impacts of food waste [13,14]. In addition to individual capacity and concerns, several authors have suggested considering opportunities such as local shopping options [15] and the availability of high-tech kitchen appliances [16]. Demographics also play a role in the generation of food waste. Such factors include household size [17,18]; household composition, including age; the presence of children; the relationship structure in the house [18]; employment status [19]; income [20]; and education level [16].

The scientific literature has explored the relationships between dietary patterns, their drivers of choice, and pro-environmental behavior, such as an increase in fruit and vegetable consumption and a decrease in meat intake [21–23]. Many studies have analyzed dietary choices in relation to personal factors such as instruction level [24–26], personal psychological factors [26–28], personal motivations, attitudes, health concerns [29–31], the availability of healthy food [32], and exposure to news and advertising related to food consumption [33]. Among the individual factors, mindfulness, considered as the ability to live in the present moment without judging the situation, seems to be relevant to the definition of dietary choices [34–36]. Finally, socio-demographic factors were found to be correlated with dietary choices [37].

In addition to individual factors, macroeconomic factors seem to influence individual dietary choices. The economic development status of countries appears to be correlated with the widespread adoption of standardized diets based on the overconsumption of meat and over-processed food [38,39]. The economic affordability of diets is also correlated with the frequency of their adoption [40,41].

Dietary choices, however, have rarely been studied as possible drivers of household food waste. Recent research has focused on the link between diets in terms of nutrient intake and pro-environmental behaviors both at aggregate [42] and consumer level [29] indirectly addressing the association with food waste [43]. Additionally, Reynolds et al. [44] showed that effective strategies for reducing food waste at the consumer level include changes in the sizes and types of plates and interventions aiming to increase the consumption of vegetables, while Qi et al. [45] highlighted the positive role of information on the social cost of food waste and its reduction. This might indicate that the shift toward healthy diets can be part of food waste reduction strategies [44]. From prior research, some promising insights seem to suggest that the research gap between dietary patterns and food waste is worth investigating because it should be considered when developing waste reduction strategies.

Building on an original survey conducted in eight countries in 2021 (Canada, China, Germany, Italy, Russia, Spain, the UK, and the USA) by Waste Watcher International, this study explored the relationship between food waste at the household level and dietary habits, and their related food choices, through a cross-country comparative perspective.

The paper is structured as follows: Section 2 outlines the survey and the methodology applied in the study; in Sections 3 and 4, the results are shown and analyzed in the frame of the existing literature; finally, Section 5 discusses the conclusions and implications for research and policy.

## 2. Materials and Methods

### 2.1. The Waste Watcher International Questionnaire

This study builds on the work of the Waste Watcher International Observatory on Food and Sustainability, which provides knowledge and tools for investigating the social, behavioral, and lifestyle dynamics behind household food waste by focusing on its economic, social, and environmental impacts. The Observatory stands to generate common and shared knowledge, to guide private and public policies for food waste prevention and reduction, and to improve food resource use efficiency. The Waste Watcher Observatory conducts research based on opinions and self-perceptions. In 2021, the Waste Watcher

International Observatory carried out a survey in eight countries—the USA, Spain, Germany, the UK, Russia, China, Canada, and Italy—to assess food waste and consumption habits through a cross-country comparative perspective. A structured questionnaire was developed by researchers involved in the Observatory, with twenty items that investigated food purchasing behaviors, food diets, and an approximate measure of waste products for the main food typologies. The full questionnaire is reported in Appendix B.

A professional market research organization, IPSOS, was contracted for the recruitment and data collection of the survey for all countries. For each country, the sample included 1000 interviews, resulting in a total of 8000. The sample was selected for each country to be representative of the adult population (18+ years old), with representative quotas for gender, age, and region. The survey was conducted online between the 13th and the 24th of August 2021 using the CAWI methodology, with a total of 15 min for each interview. All respondents provided informed consent before filling out the survey, which was compliant with the General Data Protection Regulation (GDPR).

*2.2. Defining Diets and a Measure for Household Food Waste*

Estimations of the food waste generated by the households included in the survey were based on self-reported data from respondents, who were asked to declare the quantity of food wasted during the week prior to the survey, according to 41 typologies of food. To minimize bias, respondents were asked to report indicative quantities of food wasted per typology (e.g., a cup of vegetables, the number of pieces of fruit). The amount of wasted food was then estimated considering an average value of the weights associated with the indicative measures. Out of the 41 categories, six were excluded because of their marginal contribution to the overall level of declared food waste (e.g., alcoholic and non-alcoholic beverages, fats for cooking, and sauces for seasoning). Then, the 35 remaining categories of food indicated in the questionnaire were merged into 8 macro-groups, namely: (i) fresh fruit, vegetables, and legumes (FVL); (ii) egg and dairy products; (iii) fresh seafood; (iv) frozen products; (v) fresh meat, including cured meats; (vi) processed and ready foods; (vii) starchy products, excluding potatoes; and (viii) sweet products. The total perceived quantity of food waste was calculated at the household level for each macro-group and divided by the size of the household.

To provide an estimation of the nutritional value of food wasted instead of its value in terms of weight or cost, the measure of kilocalories (Kcal) wasted in the respondents' households was computed. Adopting kilocalories as a scale to estimate the quantities of wasted food is a method widely adopted in the literature on food waste [46–49], because it overcomes the problem of aggregating diverse types of wasted food, by assessing their impact on total waste in terms of energy instead of weight, providing more generalizable results.

This approach to food waste measurement, focusing on the caloric value of wasted food, may over-represent the influence of calorie-dense foods, in comparison to nutrient-dense foods such as vegetables, fruits, seafood, and dairy products that are wasted at high rates in terms of volume. However, the impact of these products on the overall sustainability of the food system is disproportionately high due to the large number of resources needed to produce them [50–52]. This makes their possible over-representation consistent with the aim of assessing the overall impact of household food waste.

As a first step, the average quantity of Kcal per 100 g was calculated for each macro category of food. Then, these values were multiplied by the corresponding declared quantities of food waste. Estimations of the average kilocalorie values were performed using the nutritional values of foods included in a dataset elaborated by the Italian Council for Research in Agriculture and the Analysis of Agricultural Economics [53]. The dataset included an extended set of validated data on the nutritional characteristics of a wide variety of food. The average values of kilocalories per 100 g of food related to the 8 macro categories considered in this study are reported in Table 1.

**Table 1.** Quantity of Kcal per 100 g of food.

| Category of Food | Kcal (100 g) |
|---|---|
| Meat | 201.4 |
| Fish | 72.3 |
| Fruit, vegetables, and legumes, incl. potatoes (FVL) | 57.3 |
| Egg and dairy | 228.6 |
| Starchy foods (excl. potatoes) | 340.9 |
| Sweets | 247.3 |
| Ready-to-eat meals [1] | 145.1 |
| Frozen [2] | 116.4 |

Source: authors' elaboration of CREA data. Note: [1]: average of the values of Kcal/100 g of starchy foods, FLV, pizza Napoletana margherita STG (253 Kcal/100 g, provided by CREA), fast food hamburger (251 Kcal/100 g, provided by CREA), and French fries (192 Kcal/100 g, provided by CREA); [2]: average of the values of Kcal/100 g of meat, fish, and FVL.

In the questionnaire, the type of diet followed by the respondents was explored through a set of 10 statements. Respondents selected the sentence that most accurately described their prominent dietary patterns. These statements were developed to not only identify well-defined diets such as vegan and vegetarian, but also to capture the heterogeneity behind omnivorous eating regimes. From this question and its statements, five categories of diets were defined: (i) vegetarian; (ii) healthy and sustainable; (iii) traditional; (iv) smart; and (v) confused. Group (i) included individuals who self-declared as vegetarian, so they excluded meat and fish products; group (ii) consisted of individuals who declared being careful with fat intake and to prefer biological and/or local products; group (iii) encompassed respondents adhering to a Mediterranean diet or other country-specific traditional dietary patterns; group (iv) included individuals who mainly ate pre-packaged, ready-made, and convenience meals; and group (v) referred to respondents who did not identify with any particular dietary pattern and self-declared as having an irregular and confused eating regime.

Finally, 84 observations were excluded from the sample because individuals who defined themselves "vegan" had declared non-trivial meat consumption. One reason for this is that "vegan" can refer also to a lifestyle which is influenced by current trends and does not always correspond to a solid food choice. These observations were distributed independently according to age, gender, and country. The total sample was composed of 7916 observations.

### 2.3. Data Analysis

An exploratory analysis with descriptive statistics was performed. The relationship between the amount of household food waste per capita (expressed in Kcal) and the type of diet was estimated using multiple linear regression to test for correlations, with food waste as the dependent variable and the factors of each type of diet and countries as independent variables. Interactions between the country and diet factor variables were included in the model to account for heterogeneous cross-country effects of diets. Several socio-demographic variables were also integrated as controls including age, gender, education level, living arrangement (e.g., living on their own, living in couples), and a factor variable for the presence of children in the household. An alternative specification was also tested in which different regressions were adopted to estimate the differences in household food waste for each type of diet compared with all the others. The results were coherent with the main specification and are presented in Appendix A.

### 3. Results

### 3.1. Descriptive Statistics

The sample composition in terms of demographics is illustrated in Table 2. The European countries, plus the UK and Canada, were similar in terms of the mean age; this

varied between 47.8 and 49.9 years old. China and Russia presented lower mean ages (39.7 and 44.6 years, respectively), whereas the USA presented a mean age quite close the sample average.

**Table 2.** Demographic composition of the sample.

|  | Italy | Germany | China | USA | Spain | Canada | UK | Russia | Total |
|---|---|---|---|---|---|---|---|---|---|
| Mean age | 49.9 | 49.5 | 39.7 | 46.5 | 47.9 | 47.8 | 48.1 | 44.6 | 46.7 |
| Share (%) of males in the sample | 0.48 | 0.50 | 0.51 | 0.48 | 0.49 | 0.47 | 0.49 | 0.45 | 0.48 |
| Children in HH | 0.44 | 0.28 | 0.63 | 0.22 | 0.47 | 0.26 | 0.26 | 0.52 | 0.39 |
| Education: low (% share of sample) | 1.90 | 0.40 | 0.00 | 0.10 | 3.20 | 5.90 | 0.40 | 0.60 | 1.56 |
| Education: medium (% share of sample) | 64.00 | 76.10 | 22.00 | 52.60 | 40.00 | 32.50 | 62.00 | 39.20 | 48.55 |
| Education: high (% share of sample) | 34.10 | 23.50 | 78.00 | 47.30 | 56.80 | 61.60 | 37.60 | 60.20 | 49.89 |
| Living on their own | 15.30 | 33.10 | 11.60 | 30.20 | 14.40 | 30.20 | 26.20 | 14.20 | 21.90 |
| Living in a couple (w/o children) | 66.40 | 53.40 | 77.90 | 45.70 | 66.30 | 50.30 | 54.80 | 66.20 | 60.12 |
| Other (e.g., single parents, friends, roommates) | 18.30 | 13.50 | 10.50 | 14.10 | 19.30 | 19.50 | 19.00 | 19.60 | 17.97 |

The proportion of males was similar across countries, with a balanced proportion with females. On the other hand, the proportion of households with children living there (both underage and of age) was considerably different across countries: it ranged from 63% in China down to 22% in the USA. Germany, Canada, and the UK also presented a percentage of families with children living in the house of below 30%, whereas Italy, Spain, and Russia were above 40%. This clearly reflects the distinct cultural habits in terms of living standards and traditions. Notably, countries also differed regarding their living arrangements: in China, nearly 78% of the respondents lived in couples, whereas only 22% lived on their own or in other living conditions. Other countries with higher shares of people living in couples were Italy, Spain, and Russia. In contrast, in Germany, only 53% of the sample declared living in couples, whereas 33.1% declared living on their own, and 13.5% had other living arrangements.

Regarding education, most of the sample was split between medium and high education with a residual share of respondents who had only achieved a low level of education. Across countries, these shares varied considerably. The greatest proportion of people with a high education level was in China (78%); these types of surveys tend to over-represent the urban population in this country, thus supporting the comparability of populations across the selected countries.

Regarding dietary habits across countries (presented in Figure 1), more than half of the population in Italy (55.2%) and Russia (54.4%) followed a traditional type of diet, with Spain (46.1%) and China (47.1%) just below. China also presented the largest share of respondents who declared following a healthy and sustainable diet (39.5%), with Spain (30.7%), Germany (16.0%), and Italy (28.0%) closely following. Respondents self-declaring as vegetarians were more frequent in the UK (5.5%), Germany (5.0%), and Canada (4.8%). The USA (5.4%), the UK (4.1%), Canada (3.3%), and Germany (3.0%) presented higher proportions of respondents who declared following a smart diet. Similarly, Germany (42.7%), the USA (38.7%), Canada (38.4%), and the UK (29.5%) had higher shares of confused diets.

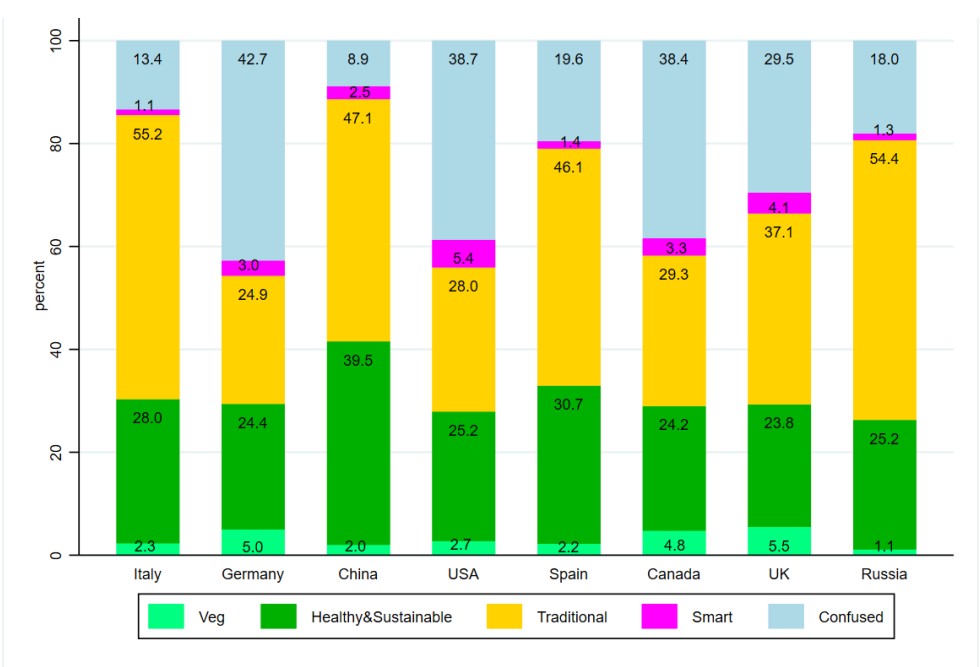

**Figure 1.** Answer to the question "How would you define your diet?", presented as percentages by country.

Analyzing food waste measured in Kcal per person (Figure 2), it was possible to notice that people following a smart diet profile were those who declared the highest value of calories wasted (4428.6). Below this, but still above the sample average, were those who declared following a vegetarian diet (2647.7). In contrast were people who declared following a traditional type of diet (1588.3). Around the sample average were individuals who followed healthy and sustainable (1952.7) and confused (1782.3) diets.

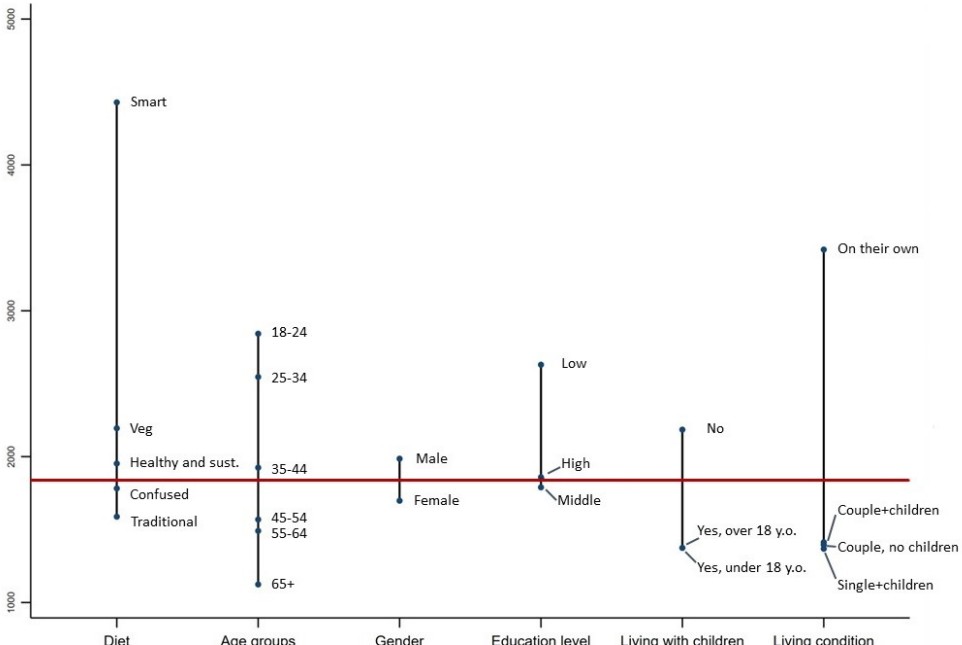

**Figure 2.** Mean food waste per person per week measured in Kcal, presented as values for all countries.

Age appeared to be negatively associated with the number of Kcal wasted; the older the cohort, the fewer kilocalories wasted (from 2886.8 to 1123.6). Of course, this might be

connected to the fact that elderly people usually need fewer calories per day [54]. According to previous results in the literature, men waste more food than women (2018.0 vs. 1711.7), as do people with a lower education level (2624.2) compared with those with medium (1823.9) and high (1871.4) levels of education [32,33]. A clear difference was also observed between people who did not live with their children (2216.2) and those who had sons/daughters living in the house (1385.8, with no significant difference regarding the age of children). According to the relationship status and/or the living arrangement, it is worth noting that people who live on their own significantly wasted more food (3475.6) than people living in couples (1412.3) or in other living/relationship arrangements, such as single parents or those living with friends or roommates (1389.7). Finally, across countries, the USA presented the highest level of food waste (2854.9), whereas Russia presented the lowest (1114.6). Canada (2213.0), China (2181.4), and Germany (2048.8) presented levels above the sample average (1860.1); the UK (1771.6), Spain (1421.0), and Italy (1275.4) presented levels below the sample average.

When analyzing the distribution of food waste (measured in kilocalories) with respect to diets and food categories (see Figure 3), it is worth noting that, across all diet profiles, the type of food most commonly wasted in terms of calories is starch. Regarding meat, vegetarian respondents declared a certain number of wasted calories from meat, even though the median was set to zero. As stated before, although the question on diet directly addressed those interviewed, the amount of food waste was measured at the household level, where people with different types of diet may live together. Additionally, regarding meat, the group that presented a distribution more prone to higher values was the smart diet group. The same also happened for the other food categories, particularly egg and dairy, ready, and sweet products, which seemed to be wasted more for this type of diet. In contrast, for people following a traditional type of diet, the distribution presented lower maximum values for almost all of the food categories. Fish and frozen products were those foods which seemed to generate less waste across all diets. Regarding FVL, when measured in weight, waste for this food category was higher for the vegetarian and healthy and sustainable types of diet. It is worth noting that these foods are low in calories, and contribute very little to global waste.

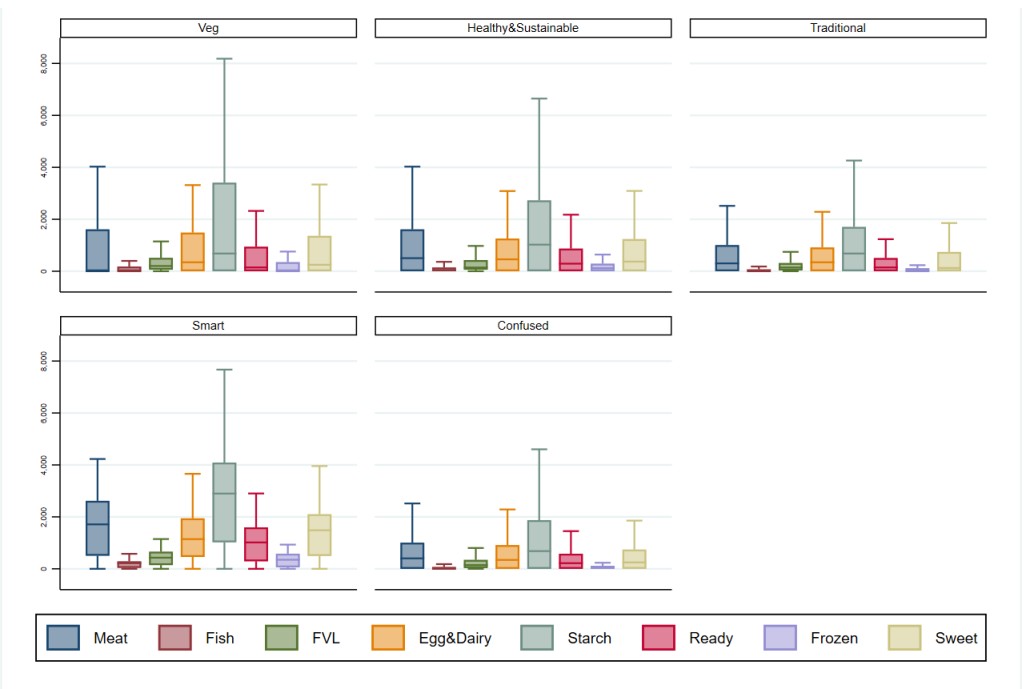

**Figure 3.** Mean food waste per person per week measured in Kcal, presented as values for all countries.

The adoption of food waste-reducing behaviors by respondents was fairly frequent among respondents from all countries, where over half of the respondents declared observing them always or often. However, some differences were identified related to the typologies of self-declared dietary habits, as reported in Table 3.

**Table 3.** Adoption of food waste reduction behavior per typology of diet (percentage of respondents declaring always and often).

| | Use of a Shopping List | Proper Storage and Use of Leftovers | Food Stock Awareness | Use of Food Products Even Shortly after Expiration Date |
|---|---|---|---|---|
| Vegetarian and Vegan | 69.1% | 78.4% | 69.9% | 73.7% |
| Healthy and Sustainable | 70.7% | 76.2% | 79.4% | 74.0% |
| Traditional | 67.7% | 75.7% | 78.4% | 73.9% |
| Smart | 58.7% | 60.1% | 70.7% | 70.7% |
| Confused | 66.5% | 74.9% | 68.6% | 78.0% |

Among food waste reduction behavior, the use of food products after the expiry date was the most frequently reported among respondents of all countries, followed by the awareness of food stock content and organization. The use of a shopping list, although still relatively widely adopted, was the least frequently adopted behavior.

Respondents who declared adopting a smart diet were the least likely to use a shopping list (58.7%), to have proper knowledge of the content and the organization of their food stock (60.1%), and to use edible food products even after the expiration date if they were still deemed safe (70.7%).

In contrast, respondents who declared adopting traditional and healthy and sustainable diets adopted behaviors that have a positive impact on food waste reduction more frequently. They presented the highest levels of awareness about the content and organization of their food stock (78.4% and 70.7%, respectively), of the proper use and storage of leftovers (75.7% and 76.2%, respectively), and of the use of food products after the expiration date if they were still deemed edible (73.9% and 75%, respectively).

*3.2. Regression Results*

The last step of data analysis was the estimation of multiple regression models, to investigate the relationship of dietary patterns with the quantity of household food waste produced by different consumer typologies, controlling for several socio-demographic factors. In particular, the regression model included factors such as country (reference value: Italy), typology of diet (reference value: traditional), age in completed years, gender (reference value: male), living arrangements (reference value: independently), education level (reference value: low), and the presence of children (reference value: yes, of age). The coefficients of the regression model are illustrated in Figure 4, with the red line indicating coefficients set to zero for the reference values. Additionally, confidence intervals at 95% are presented. The first set of regressors (country) represents the mean difference in perceived food waste for each country-specific traditional diet compared with Italy. The second group of regressors (diet) represents the mean difference in perceived food waste for each diet in Italy compared with a traditional diet. The third set of regressors (interactions) represents the mean difference between each diet in each country compared with the traditional Italian diet. Baseline coefficients are represented with zeros.

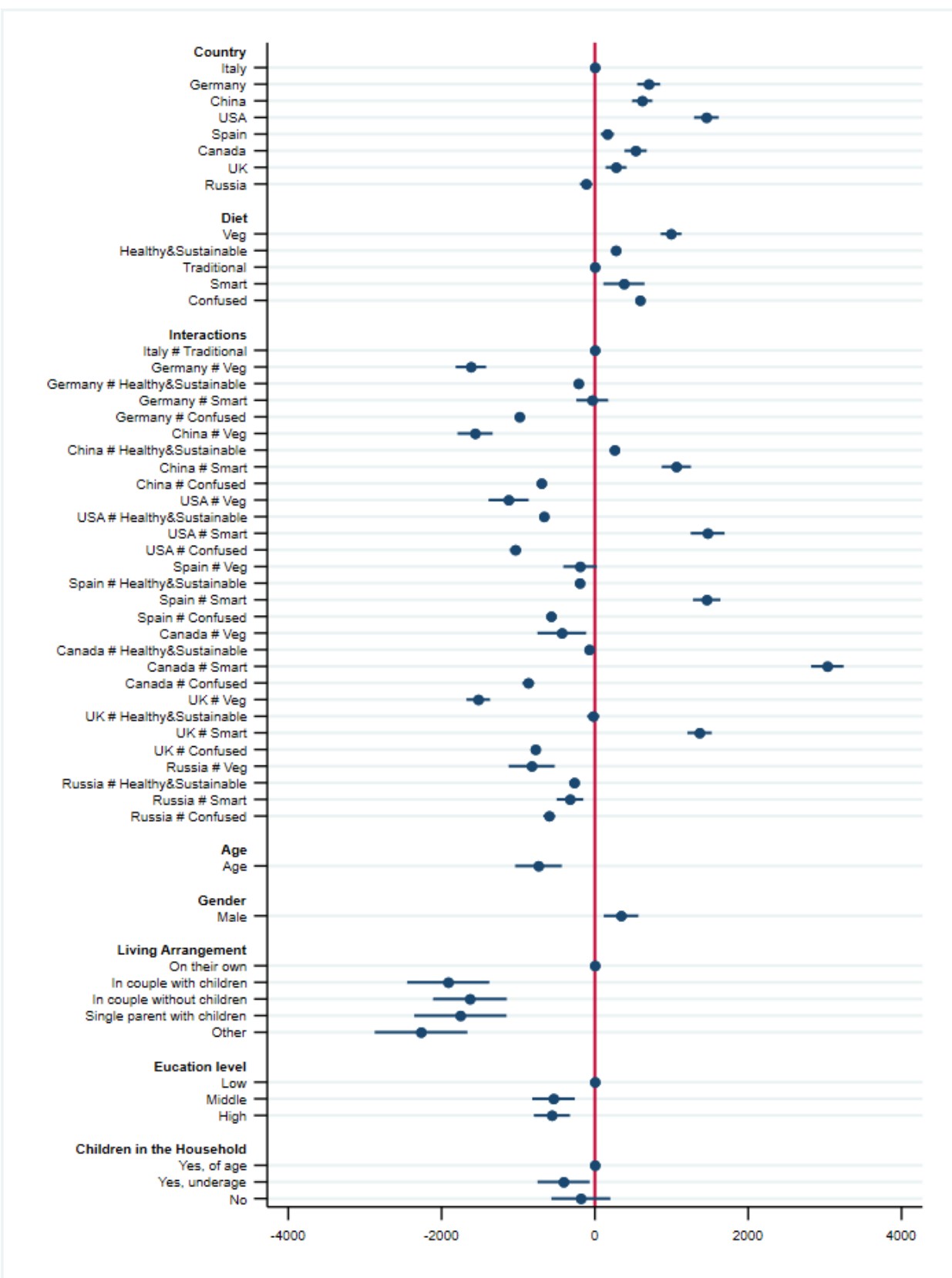

**Figure 4.** Regression coefficients for food waste (expressed in kilocalories).

Considering differences across countries regarding traditional diets (country group of coefficients in Figure 4), it is worth noting that with respect to Italy, only Russian

respondents declared wasting fewer calories than Italian consumers. Considering the traditional diets in other countries, consumers from the USA declared the highest quantities of wasted kilocalories compared with Italy, followed by Germany, China, and Canada. In Spain and the UK, the average difference from Italy between individuals who followed a traditional type of diet was minimal but still significant.

When comparing the types of diet in Italy (diet group of coefficients in Figure 4), all the coefficients associated with the five types of diet were higher on average than the coefficients representing traditional consumers, with the greatest difference for vegetarians.

The interaction coefficients between country and dietary habits represented the average differences between any type of diet in a given country and its traditional diet, compared with the same difference from the traditional Italian diet.

Consumers who declared following a smart diet tended to waste considerably more calories than consumers following other types of dietary patterns across all countries. In particular, the largest positive difference in the quantity of wasted calories was declared by Canadian respondents, followed by consumers in the USA, Spain, and the UK adopting the same dietary habits.

On the other hand, across countries, the greatest negative difference in wasted kilocalories was declared by consumers following a vegetarian diet. In particular, vegetarians from Germany declared the greatest negative difference in the amount of food wasted across the sample, followed by vegetarians from China and from the UK. Among other dietary patterns, consumers in the USA who declared a confused dietary pattern registered an amount of wasted Kcal per capita per week lower than the reference value of Italians following a traditional diet.

The controls presented coefficients coherent with the literature, because consumers with lower levels of education tended to waste more food. Consumers with secondary or higher education levels tended to waste fewer kilocalories per capita per week than the reference level of respondents with primary education or lower. Finally, the composition of the household had a statistically significant impact on the level of household food waste; households with children tended to waste more kilocalories per capita per week compared with the reference value of households with children older than 18 years old. Households with no children, on the other hand, declared higher values of wasted kilocalories compared with the other typologies of households.

## 4. Discussion

The results from the survey conducted in the eight countries highlighted significant differences among the relationships between the declared dietary patterns adopted by respondents and the self-reported amounts of food waste generated within their households.

Across countries, Russian respondents who declared following a traditional diet were associated with producing the lowest level of food waste. Italian and Spanish households of respondents adopting traditional diets followed closely. The most wasteful traditional diets in terms of wasted calories were American and the German, closely followed by traditional Chinese, Canadian, and British dietary patterns.

In Italy, individuals adopting a traditional type of diet belonged to households which, on average, wasted less food, closely followed by those who adopted a healthy and sustainable diet. In contrast, Italian respondents who declared following vegetarian, smart, and confused diets presented the highest values of self-reported household food waste.

In Germany, respondents who declared adopting vegetarian or confused dietary patterns belonged to households that, on average, wasted less food compared with those who followed a traditional or a healthy and sustainable diet. On average, more food waste was generated in the households of respondents who declared adopting a smart diet. In China, similarly to Germany, individuals who followed a vegetarian diet belonged to households which, on average, wasted less food in comparison to those who followed a traditional diet. On average, more food waste was generated in the households of respondents following a smart diet. In the USA, Spain, and Canada, in comparison with

those who followed a traditional type of diet, healthy and sustainable and confused diet followers wasted slightly less. In contrast, households of smart diet followers wasted significantly more food compared with respondents who declared different dietary habits.

In the UK, respondents following vegetarian diets were the group who declared wasting fewer calories, closely followed by Chinese and German vegetarians. Only vegetarians in Canada and the USA reported higher amounts of food waste than in Italy. Notably, the share of people following this type of diet was the lowest in all countries.

In Canada and the USA, respondents following smart diets declared the highest values of wasted calories. In the UK, Spain, Germany, and China, the self-reported level of food waste was considerably higher compared with Canada and the USA. In Italy and Russia, the differences in terms of wasted calories between smart and traditional diets were barely significant. Notably, the share of people following a smart dietary pattern was the second lowest in almost all countries.

Respondents following vegetarian diets in the UK were associated with the lowest level food waste, closely followed by Chinese and German vegetarians. Additionally, for these countries, more food waste was generated on average in the households of smart diet respondents. Consumers following a smart diet tended to waste considerably more calories than consumers following other types of dietary patterns across all countries. In particular, the largest positive difference in the quantity of wasted calories was declared by Canadian respondents, followed by consumers in the USA, Spain, and the UK adopting the same dietary habits. However, across countries, the greatest negative difference in wasted kilocalories was declared by consumers following a vegetarian diet. In particular, vegetarians from Germany declared the greatest negative difference in the amount of food wasted among all the samples, followed by vegetarians from China and from the UK. Among other dietary patterns, USA respondents declaring a confused dietary pattern were associated with a lower number of wasted kilocalories per capita per week than the reference value of Italians following a traditional diet.

In general, being a vegetarian in Germany, China, the UK, and the USA was associated with producing significantly less waste than being a vegetarian in Italy. This is possibly because in Italy, the consumption of fresh fruit and vegetables is traditionally higher than in other countries, where preserved vegetables are consumed more often. The more frequent consumption of fresh fruit and vegetables could thus generate an increase in the volume of wasted calories due to the greater perishability of this type of product, leading to more frequent disposal.

In Italy, traditional and healthy and sustainable diets were associated with the lowest self-declared amount of food waste, whereas vegetarian and smart and confused diets were associated with more food waste per capita in terms of kilocalories. Across countries, traditional Russian and Spanish traditional diets were also associated with households reporting low levels of food waste. The most wasteful traditional diets in terms of calories were American and German, closely followed by the traditional Chinese, Canadian, and British diets.

Given these differences and patterns, it is worth noting that consumers following diets associated with lower levels of declared food waste consume more raw products and less prepared/processed food than consumers following diets associated with higher levels of food waste. Additionally, the use of leftovers and overbuying were among food-related behaviors that differed the most between dietary patterns. Consumers' habits that were considered more important in avoiding food waste were following diets correlated with lower levels of food waste, the use of shopping lists, the proper use of leftovers (e.g., freezing, using before going bad), knowledge of food stocks, and the use of food beyond expiration dates if they are still perceived as good.

Finally, the results of this study highlight that food preparation and management skills, particularly the availability of time dedicated to preparing food, have a stronger link with lower levels of food waste declared by households, when compared with the impact of motivations and attitudes. This is consistent with the most recent literature on

household food waste drivers, which evidences a prominent role of time availability as a driver for reductions in food waste at the household level [55–58].

*Limitations*

This study was built on a consumer survey in which respondents self-reported their behavior, motivation, and amounts of food wasted. Therefore, this methodology potentially suffers from cognitive biases, such as social comparison and social desirability bias, correlated to the country. Indeed, 9% of the whole sample declared not having produced food waste in the week before the interview; this distribution was not independent across countries or age categories (chi-squared test). Although these weaknesses of surveys have been well-recognized and discussed in the literature, the CAWI method remains a solid method to explore food waste and its related behaviors [59].

A second limitation was related to the definition of dietary patterns which, in some cases, could present some overlapping (e.g., Mediterranean and traditional diets were similar for southern European countries), due to the variety of contexts considered in this study. However, these cases were limited and did not affect the overall consistency of the results.

A final limitation derives from the fact that questions related to dietary habits were designed for an individual response, whereas self-reported measures of food waste referred to the whole household. However, because the questionnaire was compiled by individuals who were responsible, or at least partially responsible, for the management of the food in their households, it is possible to affirm that their dietary pattern could be a proxy for the main dietary pattern of the household.

## 5. Conclusions

Dietary patterns have rarely been studied as possible drivers of household food waste, although they could represent a valuable element of food waste reduction strategies. Building upon an original cross-country survey conducted in 2021 by Waste Watcher International, this study explored the relationship between food waste at household level and dietary habits, and their related food choices, through a cross-country comparative perspective.

Several conclusions can be drawn from this study. First, culinary traditions have a role in shaping consumption decisions, and consequently, they also impact perceived food waste production. However, traditional diets are not uniquely associated with lower or higher levels of food waste, meaning that new and emerging dietary habits are not necessarily associated with higher levels of food waste. In more detail, the results show that smart diets are frequently associated with the highest values of food waste. This was valid for all countries, although especially Canada, Spain, the UK, and the USA. Vegetarian diets were associated with lower values of food waste, being based on foods with a low density of calories, especially when compared with traditional diets in China, Germany, the UK, and the USA. This was not true for Italy, where the traditional diet, followed in most Mediterranean countries, is associated with lower self-declared amounts of food waste. Other traditional diets associated with less food waste were Russian and Spanish.

Second, lower levels of food waste were associated with types of diet which usually require more raw products, more time dedicated to food preparation, and greater kitchen skills. Indeed, a considerable difference was present between diets regarding the possible use of leftovers. In contrast, higher levels of food waste were associated with the types of diet that relied on more processed and ready-to-eat food. These types of foods are generally high in fats and sugars, and are sold in portions that might be too big for consumers and generally are difficult to reuse as leftovers. Indeed, it appears that being engaged with food consumption concerning local and seasonal products or utilizing convenience, highly processed, and ready-to-eat foods may be associated with different self-reported levels of food waste. This reinforces the concept that food waste prevention strategies have more to

do with skills, time, and the management of food, especially leftovers, than motivations and attitudes.

Third, the proportion of individuals following different types of diet was fairly uneven. Countries with more respondents for traditional diets were coherent with the places that had stronger cultural food traditions, whereas confused diets were more prominent in countries with less peculiar culinary traditions. Notably, the share of the population that declared following a smart diet was approximately 2.7% of the whole sample. Consumers following this type of dietary pattern seemed to be mostly responsible for the larger share of perceived food waste generation; therefore, it could be worth developing food waste prevention interventions that target specific dietary patterns and their associated food choices.

These types of interventions could also generate positive spillover, supporting the transition towards diets that make less use of pre-packaged, ready-to-eat meals. Improving diet quality could, indeed, simultaneously reduce the environmental impact of food waste and unhealthy and unsustainable food choices.

**Author Contributions:** Conceptualization, E.I. and M.M.; methodology, E.I.; validation, E.R. and A.S.; formal analysis, E.I. and M.M.; data curation, E.R. and L.F.; visualization, E.I. and M.M.; writing—original draft preparation, E.I., M.M. and M.V.; writing—review and editing, L.F., E.R. and A.S.; supervision, M.V. and A.S. All authors have read and agreed to the published version of the manuscript.

**Funding:** This research received no external funding.

**Data Availability Statement:** Data can be provided by Authors on request.

**Conflicts of Interest:** The authors declare no conflict of interest.

**Appendix A. Regression Results**

Table A1 presents the specification of the model in which food waste was tested against all typologies of diet. The first column presents the main model specification presented in Figure 4. In the other columns, the dependent variable is decomposed into each food category to account for different types of waste produced in each context. For each model, the first set of regressors represents the mean difference in perceived food waste for each country-specific traditional diet compared with Italy. The second group of regressors represents the mean difference in perceived food waste for each diet in Italy compared with the traditional diet. The third set of regressors represents the mean difference between each diet in each country compared with the traditional Italian diet. Baseline coefficients are represented with zeros.

Table A2 presents an alternative model specification where food waste is regressed considering each diet independently, compared with all the others. For each model, a dummy variable has a value of 1 for each type of diet, and 0 for all the others. The interaction between the diet dummy variable and the set of country-related dummy variables (in bold), represents the mean difference for each type of diet in each country compared with Italy.

**Table A1.** Main model specifications with food waste regressed against all diets.

| | MAIN MODEL | | Meat | | Fish | | FVL | | Egg and Dairy | | Starch | |
|---|---|---|---|---|---|---|---|---|---|---|---|---|
| Baseline = Italy | | | | | | | | | | | | |
| Germany | 754.6 *** | (21.67) | 146.5 *** | (20.69) | 3.725 *** | (5.72) | 41.79 *** | (26.67) | 134.9 *** | (27.98) | 258.9 *** | (25.11) |
| China | 642.8 *** | (16.64) | 144.3 *** | (18.09) | 16.11 *** | (27.26) | 27.30 *** | (13.91) | 68.24 *** | (10.59) | 184.1 *** | (16.20) |
| USA | 1499.5 *** | (40.71) | 277.8 *** | (44.34) | 22.89 *** | (33.62) | 59.43 *** | (34.69) | 216.1 *** | (34.52) | 467.4 *** | (44.02) |
| Spain | 199.1 *** | (17.07) | 46.88 *** | (14.28) | 5.990 *** | (22.30) | 9.116 *** | (12.63) | 18.41 *** | (14.63) | 50.37 *** | (12.64) |
| Canada | 568.7 *** | (21.34) | 89.84 *** | (21.53) | 8.887 *** | (14.91) | 32.54 *** | (23.28) | 76.19 *** | (16.46) | 195.0 *** | (25.83) |
| UK | 323.0 *** | (14.67) | 66.84 *** | (17.96) | 0.218 | (0.47) | 26.51 *** | (26.96) | 51.57 *** | (14.67) | 113.4 *** | (17.73) |
| Russia | −77.89 * | (−3.48) | −16.68 * | (−3.26) | −5.704 *** | (−13.62) | 2.184 | (2.06) | −8.038 * | (−2.79) | −11.63 | (−1.63) |
| Baseline = Traditional | | | | | | | | | | | | |
| Veg | 1244.7 *** | (20.07) | 32.01 * | (3.38) | 36.35 *** | (21.70) | 78.04 *** | (29.17) | 217.4 *** | (21.04) | 497.8 *** | (28.68) |
| Healthy and Sustainable | 168.8 *** | (11.31) | 24.75 *** | (9.85) | 0.316 | (1.13) | 5.427 *** | (6.91) | 29.00 *** | (9.81) | 42.31 *** | (8.73) |
| Smart | 147.9 | (1.84) | 142.2 *** | (10.26) | 13.12 *** | (8.65) | 7.942 | (2.02) | −62.46 ** | (−4.91) | −48.30 | (−1.98) |
| Confused | 398.1 *** | (12.13) | 55.03 *** | (9.18) | 6.182 *** | (8.72) | 13.29 *** | (9.76) | 53.17 *** | (10.79) | 112.1 *** | (11.05) |
| Germany # Veg | −1877.3 *** | (−27.20) | −291.3 *** | (−24.04) | −45.67 *** | (−27.08) | −96.36 *** | (−37.41) | −269.5 *** | (−27.41) | −665.7 *** | (−32.36) |
| Germany # Healthy and Sustainable | −113.3 *** | (−6.15) | −42.47 *** | (−13.93) | 6.485 *** | (20.98) | 4.926 ** | (5.36) | 1.899 | (0.59) | −51.30 *** | (−9.26) |
| Germany # Smart | 198.0 ** | (3.71) | −85.42 *** | (−8.19) | 7.684 *** | (5.97) | −10.66 ** | (−4.41) | 15.82 | (2.05) | 8.493 | (0.52) |
| Germany # Confused | −788.9 *** | (−28.81) | −155.6 *** | (−28.64) | −9.692 *** | (−14.58) | −30.54 *** | (−32.07) | −100.5 *** | (−23.67) | −272.7 *** | (−33.29) |
| China # Veg | −1824.2 *** | (−19.17) | −146.6 *** | (−9.44) | −47.33 *** | (−19.30) | −110.3 *** | (−30.18) | −321.4 *** | (−22.39) | −637.9 *** | (−23.37) |
| China # Healthy and Sustainable | 360.6 *** | (29.07) | 72.05 *** | (28.12) | 9.935 *** | (28.92) | 13.87 *** | (25.98) | 46.70 *** | (32.28) | 127.3 *** | (32.80) |
| China # Smart | 1285.5 *** | (26.85) | 90.58 *** | (10.96) | 21.28 *** | (23.22) | 57.56 *** | (22.82) | 287.5 *** | (39.08) | 447.6 *** | (31.26) |
| China # Confused | −514.4 *** | (−51.61) | −66.15 *** | (−29.88) | −3.720 *** | (−15.13) | −14.68 *** | (−30.41) | −100.7 *** | (−87.06) | −165.6 *** | (−49.68) |
| USA # Veg | −1381.8 *** | (−23.52) | −126.5 *** | (−12.19) | −32.17 *** | (−21.26) | −17.49 *** | (−7.26) | −174.5 *** | (−22.42) | −567.9 *** | (−30.49) |
| USA # Healthy and Sustainable | −558.7 *** | (−46.86) | −76.39 *** | (−39.87) | 1.448 ** | (5.26) | −19.83 *** | (−31.51) | −66.81 *** | (−29.83) | −199.4 *** | (−49.10) |
| USA # Smart | 1709.8 *** | (55.54) | 168.1 *** | (25.92) | 67.66 *** | (75.90) | 66.98 *** | (41.31) | 258.9 *** | (49.52) | 681.6 *** | (83.67) |
| USA # Confused | −848.0 *** | (−21.90) | −142.4 *** | (−19.33) | −15.16 *** | (−19.66) | −25.55 *** | (−17.32) | −108.4 *** | (−17.48) | −274.4 *** | (−23.53) |
| Spain # Veg | −456.2 *** | (−8.53) | 71.44 *** | (7.23) | −28.36 *** | (−18.44) | −63.57 *** | (−30.81) | −142.9 *** | (−20.00) | −168.6 *** | (−10.38) |
| Spain # Healthy and Sustainable | −91.56 ** | (−4.56) | −24.35 *** | (−6.44) | 1.909 ** | (4.15) | 6.602 *** | (9.12) | −8.774 * | (−2.68) | −11.88 | (−1.92) |
| Spain # Smart | 1681.4 *** | (76.09) | 111.1 *** | (29.82) | 40.77 *** | (94.47) | 112.1 *** | (85.30) | 341.1 *** | (84.88) | 588.5 *** | (86.76) |
| Spain # Confused | −382.2 *** | (−28.81) | −60.53 *** | (−22.60) | −5.040 *** | (−21.17) | −2.112 ** | (−4.19) | −60.63 *** | (−23.88) | −88.29 *** | (−23.31) |
| Canada # Veg | −674.9 *** | (−8.37) | 46.08 * | (3.03) | −22.16 *** | (−11.65) | −17.05 *** | (−6.10) | −122.0 *** | (−10.84) | −369.9 *** | (−14.99) |
| Canada # Healthy and Sustainable | 37.93 | (1.78) | 25.51 *** | (6.06) | 7.745 *** | (15.24) | 7.125 *** | (7.76) | 18.18 ** | (5.34) | −12.32 | (−1.83) |
| Canada # Smart | 3263.4 *** | (143.10) | 531.1 *** | (106.65) | 68.29 *** | (92.72) | 126.2 *** | (103.46) | 465.8 *** | (122.62) | 1044.9 *** | (163.95) |
| Canada # Confused | −674.4 *** | (−17.71) | −108.2 *** | (−15.63) | −14.80 *** | (−17.43) | −30.24 *** | (−19.25) | −92.13 *** | (−16.58) | −181.2 *** | (−15.46) |
| UK # Veg | −1779.4 *** | (−27.75) | −184.1 *** | (−17.63) | −53.50 *** | (−30.78) | −82.86 *** | (−32.04) | −290.8 *** | (−28.70) | −658.8 *** | (−36.83) |
| UK # Healthy and Sustainable | 74.38 ** | (4.61) | 20.39 *** | (6.44) | 7.273 *** | (17.95) | 10.38 *** | (20.57) | 28.33 *** | (11.69) | 18.22 ** | (3.69) |
| UK # Smart | 1589.6 *** | (66.96) | 226.3 *** | (51.35) | 32.21 *** | (45.55) | 59.27 *** | (46.37) | 288.4 *** | (70.54) | 542.6 *** | (76.50) |
| UK # Confused | −583.9 *** | (−44.40) | −104.0 *** | (−38.01) | −11.37 *** | (−36.34) | −16.28 *** | (−38.64) | −67.31 *** | (−33.29) | −158.2 *** | (−40.28) |
| Russia # Veg | −1062.8 *** | (−13.55) | 65.98 ** | (4.21) | −22.74 *** | (−12.54) | −94.65 *** | (−31.99) | −153.4 *** | (−14.70) | −569.1 *** | (−23.74) |
| Russia # Healthy and Sustainable | −163.0 *** | (−10.88) | −3.947 | (−1.06) | 0.145 | (0.52) | −11.85 *** | (−15.37) | −42.82 *** | (−29.28) | −46.50 *** | (−9.59) |
| Russia # Smart | −97.33 | (−2.34) | −97.89 *** | (−11.30) | 1.826 | (1.99) | −41.03 *** | (−27.16) | 48.43 *** | (7.74) | 66.49 ** | (5.26) |
| Russia # Confused | −413.6 *** | (−20.37) | −73.59 *** | (−19.91) | −5.640 *** | (−13.81) | −14.09 *** | (−15.33) | −63.83 *** | (−20.87) | −83.70 *** | (−12.69) |

**Table A1.** *Cont.*

| | MAIN MODEL | | Meat | | Fish | | FVL | | Egg and Dairy | | Starch | |
|---|---|---|---|---|---|---|---|---|---|---|---|---|
| Age | −772.3 *** | (−6.59) | −136.1 *** | (−6.53) | −16.86 *** | (−6.98) | −28.83 *** | (−5.67) | −114.4 *** | (−6.16) | −238.1 *** | (−6.41) |
| Male | 343.7 ** | (3.80) | 62.12 ** | (3.53) | 9.256 ** | (4.24) | 10.25 * | (2.95) | 43.95 * | (2.94) | 105.7 ** | (4.07) |
| In couple with children | −1852.1 *** | (−8.47) | −316.9 *** | (−7.27) | −34.87 *** | (−8.55) | −106.5 *** | (−9.60) | −288.5 *** | (−7.74) | −570.1 *** | (−9.02) |
| In couple without children | −1562.9 *** | (−7.98) | −271.8 *** | (−7.28) | −29.91 *** | (−6.56) | −87.15 *** | (−9.79) | −241.6 *** | (−8.15) | −484.4 *** | (−8.84) |
| Single parent with children | −1691.0 *** | (−6.94) | −287.7 *** | (−6.28) | −32.08 *** | (−8.11) | −96.57 *** | (−7.18) | −256.5 *** | (−5.90) | −529.1 *** | (−8.14) |
| Other | −2220.8 *** | (−9.14) | −388.3 *** | (−8.35) | −43.53 *** | (−8.41) | −118.9 *** | (−11.53) | −336.9 *** | (−9.05) | −697.2 *** | (−9.90) |
| Middle Education | −622.6 ** | (−4.42) | −123.8 * | (−3.49) | −12.13 ** | (−4.95) | −17.09 | (−2.32) | −68.13 * | (−2.69) | −181.8 ** | (−4.05) |
| Higher Education | −642.1 *** | (−5.70) | −132.8 ** | (−4.69) | −13.35 *** | (−8.18) | −15.09 * | (−2.43) | −65.64 * | (−2.38) | −187.3 ** | (−5.27) |
| Children in HH, underage | −437.4 * | (−3.45) | −72.00 ** | (−3.81) | −10.14 ** | (−3.82) | −16.59 * | (−2.61) | −69.23 ** | (−3.52) | −135.6 * | (−3.41) |
| Children in HH No | −200.3 | (−1.24) | −30.79 | (−1.07) | −4.080 | (−1.29) | −6.571 | (−0.84) | −34.71 | (−1.19) | −54.70 | (−1.09) |
| Constant | 5055.2 *** | (11.95) | 894.4 *** | (12.13) | 103.0 *** | (13.85) | 236.6 *** | (11.60) | 742.4 *** | (9.76) | 1568.7 *** | (11.84) |
| Observations | 7916 | | 7916 | | 7916 | | 7916 | | 7916 | | 7916 | |
| R-squared | 0.199 | | 0.177 | | 0.145 | | 0.186 | | 0.180 | | 0.170 | |

t statistics in parentheses * $p < 0.05$, ** $p < 0.01$, *** $p < 0.001$.

**Table A2.** Alternative model specification with food waste regressed considering each diet independently, compared with all the others.

| | TYPE OF DIET Vegetarian | | TYPE OF DIET Healthy and Sustainable | | TYPE OF DIET Traditional | | TYPE OF DIET Smart | | TYPE OF DIET Confused | |
|---|---|---|---|---|---|---|---|---|---|---|
| Baseline = Italy | | | | | | | | | | |
| Germany | 491.2 *** | (13.49) | 377.8 *** | (9.60) | 222.7 *** | (7.48) | 433.8 *** | (14.24) | 639.9 *** | (17.55) |
| China | 752.7 *** | (18.46) | 526.8 *** | (13.36) | 748.8 *** | (18.37) | 692.4 *** | (17.27) | 794.5 *** | (20.62) |
| USA | 1208.9 *** | (34.72) | 1265.7 *** | (36.49) | 955.1 *** | (32.71) | 1078.9 *** | (40.65) | 1390.0 *** | (29.23) |
| Spain | 145.7 *** | (13.87) | 146.7 *** | (12.08) | 36.42 * | (2.59) | 112.1 *** | (9.63) | 192.3 *** | (14.78) |
| Canada | 516.1 *** | (16.09) | 483.6 *** | (13.72) | 388.0 *** | (10.86) | 408.1 *** | (13.78) | 776.1 *** | (21.88) |
| UK | 290.1 *** | (10.13) | 181.7 *** | (7.55) | 94.80 ** | (3.60) | 166.6 *** | (7.40) | 365.5 *** | (12.94) |
| Russia | −190.4 *** | (−9.52) | −201.6 *** | (−8.79) | −373.5 *** | (−14.70) | −210.6 *** | (−10.09) | −164.6 *** | (−8.58) |
| [TYPE OF DIET] == 1 | 1132.1 *** | (17.78) | 49.95 ** | (5.00) | −286.7 *** | (−13.32) | 23.61 | (0.34) | 303.3 *** | (10.48) |

**Table A2.** *Cont.*

| | TYPE OF DIET Vegetarian | | TYPE OF DIET Healthy and Sustainable | | TYPE OF DIET Traditional | | TYPE OF DIET Smart | | TYPE OF DIET Confused | |
|---|---|---|---|---|---|---|---|---|---|---|
| **Germany # TYPE OF DIET] == 1** | −1614.7 *** | (−22.81) | 258.4 *** | (13.03) | 522.4 *** | (37.96) | 507.0 *** | (9.16) | −680.3 *** | (−21.25) |
| **China # TYPE OF DIET] == 1** | −1943.5 *** | (−20.21) | 451.6 *** | (38.99) | −128.6 *** | (−13.89) | 1212.8 *** | (24.76) | −681.3 *** | (−55.91) |
| **USA # TYPE OF DIET] == 1** | −1118.1 *** | (−18.28) | −341.7 *** | (−24.00) | 532.5 *** | (35.13) | 2114.6 *** | (51.77) | −738.7 *** | (−15.55) |
| **Spain # TYPE OF DIET] == 1** | −415.7 *** | (−7.06) | −42.99 | (−1.89) | 159.9 *** | (12.69) | 1760.9 *** | (95.50) | −372.2 *** | (−30.16) |
| **Canada # TYPE OF DIET] == 1** | −658.1 *** | (−7.45) | 109.4 *** | (6.27) | 171.7 *** | (9.21) | 3408.7 *** | (156.52) | −884.4 *** | (−22.91) |
| **UK # TYPE OF DIET] == 1** | −1738.2 *** | (−24.60) | 201.0 *** | (13.40) | 224.7 *** | (26.27) | 1734.9 *** | (85.58) | −629.2 *** | (−27.26) |
| **Russia # TYPE OF DIET] == 1** | −966.7 *** | (−13.61) | −46.26 ** | (−3.75) | 290.8 *** | (14.96) | 22.77 | (0.67) | −332.3 *** | (−18.96) |
| Age | −811.2 *** | (−6.74) | −810.3 *** | (−6.73) | −806.5 *** | (−6.56) | −779.5 *** | (−6.76) | −807.5 *** | (−6.89) |
| Male | 357.5 ** | (3.89) | 360.6 ** | (3.89) | 357.8 ** | (3.88) | 345.1 ** | (3.76) | 353.0 ** | (3.72) |
| In couple with children | −1917.0 *** | (−8.12) | −1931.1 *** | (−8.16) | −1919.2 *** | (−8.08) | −1857.9 *** | (−8.71) | −1923.2 *** | (−8.37) |
| In couple without children | −1634.1 *** | (−7.66) | −1640.5 *** | (−7.71) | −1640.0 *** | (−7.60) | −1581.8 *** | (−8.39) | −1637.7 *** | (−8.05) |
| Single parent with children | −1802.5 *** | (−7.30) | −1807.7 *** | (−7.28) | −1818.4 *** | (−7.39) | −1727.3 *** | (−7.64) | −1794.0 *** | (−7.34) |
| Other | −2321.5 *** | (−8.51) | −2328.3 *** | (−8.52) | −2329.7 *** | (−8.68) | −2261.1 *** | (−9.15) | −2312.1 *** | (−9.01) |
| Middle education | −612.3 ** | (−4.21) | −593.8 ** | (−4.12) | −591.5 ** | (−4.14) | −603.6 ** | (−4.34) | −619.4 ** | (−4.26) |
| Higher education | −619.9 ** | (−4.88) | −594.1 ** | (−4.88) | −597.8 ** | (−4.91) | −613.1 *** | (−5.50) | −637.6 *** | (−5.57) |
| Children in HH, underage | −477.4 ** | (−3.55) | −487.9 ** | (−3.66) | −488.5 ** | (−3.61) | −454.1 ** | (−3.68) | −479.3 ** | (−3.66) |
| Children in HH No | −245.6 | (−1.57) | −245.1 | (−1.54) | −247.3 | (−1.58) | −219.5 | (−1.42) | −236.4 | (−1.50) |
| Constant | 5323.8 *** | (12.63) | 5322.8 *** | (12.46) | 5484.7 *** | (13.14) | 5207.5 *** | (12.92) | 5311.2 *** | (12.89) |
| Observations | 7916 | | 7916 | | 7916 | | 7916 | | 7916 | |
| R-squared | 0.181 | | 0.181 | | 0.181 | | 0.193 | | 0.182 | |

t statistics in parentheses * *p* < 0.05, ** *p* < 0.01, *** *p* < 0.001.

**Appendix B. Survey Questionnaire**

Please indicate your date of birth.
Year

- 1910
- ...
- 2015

Month

- January
- ...
- December

Are you ... ?

- Man
- Woman

Residential address?

- Province:
- Municipality:
- ZIP code:

How many people does your households consist of? (Include yourself and other people, adults or children, who have been living at your current address for at least two months)

- 1
- 2
- ...
- 12+

How is your household composed?

- Lives alone 18–34 years old
- Lives alone 35–54 years old
- Lives alone +55 years old
- 18–34 years old couple with children
- 18–34 years old couple without children
- 35–54 years old couple with children
- 35–54 years old couple without children
- Over 54 years old couple with children
- Over 54 years old couple without children
- Single parent, with children
- Single, lives with other people (friends, relatives)
- Other/prefer not to answer

Do you have children living with you?

- Yes over 18 years old
- Yes under 18 years old
- No

Q1—We would like to talk about your eating habits. Can you tell us, in your, typical week, what food do you consume?

| | Everyday | 4–5 times a week | 2–3 times a week | Once a week | A few times in a month | Rarely | Never | I don't know |
|---|---|---|---|---|---|---|---|---|
| 1. Cold cuts/salami/cured meat | ○ | ○ | ○ | ○ | ○ | ○ | ○ | ○ |
| 2. Soft drinks (fruit juices, Coke, mineral water, etc.) | ○ | ○ | ○ | ○ | ○ | ○ | ○ | ○ |
| 3. Alcoholic beverages (wine, beer, etc.) | ○ | ○ | ○ | ○ | ○ | ○ | ○ | ○ |
| 4. Butter, margarine, and oil | ○ | ○ | ○ | ○ | ○ | ○ | ○ | ○ |
| 5. Cooked WHITE meat (excluding cured meat) | ○ | ○ | ○ | ○ | ○ | ○ | ○ | ○ |
| 6. Raw WHITE meat (excluding cured meat) | ○ | ○ | ○ | ○ | ○ | ○ | ○ | ○ |
| 7. Cooked RED meat (excluding cured meat) | ○ | ○ | ○ | ○ | ○ | ○ | ○ | ○ |
| 8. Raw RED meat (excluding cured meat) | ○ | ○ | ○ | ○ | ○ | ○ | ○ | ○ |
| 9. Prepared or precooked food (portions, roast chickens, pizzas, etc.) | ○ | ○ | ○ | ○ | ○ | ○ | ○ | ○ |
| 10. Onions, garlic and tubers (potatoes, carrots, etc.) | ○ | ○ | ○ | ○ | ○ | ○ | ○ | ○ |
| 11. Sweets (cakes, ice-cream, etc.) | ○ | ○ | ○ | ○ | ○ | ○ | ○ | ○ |
| 12. Chocolate, spreads, etc. | ○ | ○ | ○ | ○ | ○ | ○ | ○ | ○ |
| 13. Cheese | ○ | ○ | ○ | ○ | ○ | ○ | ○ | ○ |
| 14. Fresh fruit | ○ | ○ | ○ | ○ | ○ | ○ | ○ | ○ |
| 15. Non-fresh fruit and vegetables (jarred, canned) | ○ | ○ | ○ | ○ | ○ | ○ | ○ | ○ |
| 16. Salads | ○ | ○ | ○ | ○ | ○ | ○ | ○ | ○ |
| 17. Milk and yogurt | ○ | ○ | ○ | ○ | ○ | ○ | ○ | ○ |
| 18. Dairy products (mozzarella, cottage cheese … ) | ○ | ○ | ○ | ○ | ○ | ○ | ○ | ○ |
| 19. Legumes (lentils, beans. chickpeas, etc.) | ○ | ○ | ○ | ○ | ○ | ○ | ○ | ○ |
| 20. Jams and marmalades | ○ | ○ | ○ | ○ | ○ | ○ | ○ | ○ |
| 21. Mayonnaise and egg-based sauces (es. Tartar, Bernese … ) | ○ | ○ | ○ | ○ | ○ | ○ | ○ | ○ |
| 22. Fresh bread | ○ | ○ | ○ | ○ | ○ | ○ | ○ | ○ |
| 23. Packaged bread | ○ | ○ | ○ | ○ | ○ | ○ | ○ | ○ |
| 24. Pasta and fresh pasta (raw) | ○ | ○ | ○ | ○ | ○ | ○ | ○ | ○ |
| 25. Cooked pasta | ○ | ○ | ○ | ○ | ○ | ○ | ○ | ○ |
| 26. Raw fish/crustaceans/shellfish | ○ | ○ | ○ | ○ | ○ | ○ | ○ | ○ |
| 27. Cooked fish/crustaceans/shellfish | ○ | ○ | ○ | ○ | ○ | ○ | ○ | ○ |

| | Everyday | 4–5 times a week | 2–3 times a week | Once a week | A few times in a month | Rarely | Never | I don't know |
|---|---|---|---|---|---|---|---|---|
| 28. Breakfast products (cookies, cereals, rusks, etc.) | ○ | ○ | ○ | ○ | ○ | ○ | ○ | ○ |
| 29. Frozen products (veggie soups, etc.) | ○ | ○ | ○ | ○ | ○ | ○ | ○ | ○ |
| 30. Rice and other cooked grains | ○ | ○ | ○ | ○ | ○ | ○ | ○ | ○ |
| 31. Rice and other uncooked grains | ○ | ○ | ○ | ○ | ○ | ○ | ○ | ○ |
| 32. Sauces (e.g., Ketchup, tabasco … ) | ○ | ○ | ○ | ○ | ○ | ○ | ○ | ○ |
| 33. Sauces (es. tomato sauce, ready-made sauces, pesto … ) | ○ | ○ | ○ | ○ | ○ | ○ | ○ | ○ |
| 34. Eggs | ○ | ○ | ○ | ○ | ○ | ○ | ○ | ○ |
| 35. Fresh vegetables | ○ | ○ | ○ | ○ | ○ | ○ | ○ | ○ |
| 36. Pizza | ○ | ○ | ○ | ○ | ○ | ○ | ○ | ○ |
| 37. Stuffed sandwiches | ○ | ○ | ○ | ○ | ○ | ○ | ○ | ○ |
| 38. Frankfurters | ○ | ○ | ○ | ○ | ○ | ○ | ○ | ○ |
| 39. French fries | ○ | ○ | ○ | ○ | ○ | ○ | ○ | ○ |
| 40. Frozen or deep-frozen products | ○ | ○ | ○ | ○ | ○ | ○ | ○ | ○ |
| 41. Prepared and precooked meals | ○ | ○ | ○ | ○ | ○ | ○ | ○ | ○ |

Q2—How would you define your diet

- Vegan
- Vegetarian
- Healthy and low-fat
- Mediterranean style, with pasta and pizza
- Sustainability-aware with organic products
- Territory-aware with local products
- Traditional diet, typical of my country
- Smart, with pre-packaged meals
- Confused and irregular, with no particular preferences
- don't know

Q3—Please select which of the following statements you most identify with:

- High quality food is very, important, and I am willing to spend to be assured of quality
- I am very pragmatic about food: I buy considering the price I think is right
- I have other priorities than food, I try to spend as little as possible
- I don't know

Q4—How often do you throw away cooked/prepared leftovers or food that you no longer consider good?

1. Almost every day
2. 3–4 times a week
3. 1–2 times a week
4. Less than once a week
5. Almost never
6. I don't know

Q5—What kind of food would you say that you throw away most often? (5 possible answers)

1. Cold cuts/salami/cured meat
2. Soft drinks (fruit juices, Coke, mineral water, etc.)

3. Alcoholic beverages (wine, beer, etc.)
4. Butter, margarine and oil
5. Cooked WHITE meat (excluding cured meat)
6. Raw WHITE meat (excluding cured meat)
7. Cooked RED meat (excluding cured meat)
8. Raw RED meat (excluding cured meat)
9. Prepared or precooked food (portions, roast chickens, pizzas, etc.)
10. Onions, garlic and tubers (potatoes, carrots, etc.)
11. Sweets (cakes, ice-cream, etc.)
12. Chocolate, spreads, etc.
13. Cheese
14. Fresh fruit
15. Non-fresh fruit and vegetables (jarred, canned)
16. Salads
17. Milk and yogurt
18. Dairy products (mozzarella, cottage cheese . . . )
19. Legumes (lentils, beans. chickpeas, etc.)
20. Jams and marmalades
21. Mayonnaise and egg-based sauces (es. Tartar, Bernese . . . )
22. Fresh bread
23. Packaged bread
24. Pasta and fresh pasta (raw)
25. Cooked pasta
26. Raw fish/crustaceans/shellfish
27. Cooked fish/crustaceans/shellfish
28. Breakfast products (cookies, cereals, rusks, etc.)
29. Frozen products (veggie soups, etc.)
30. Rice and other cooked grains
31. Rice and other uncooked grains
32. Sauces (e.g., Ketchup, tabasco . . . )
33. Sauces (es. tomato sauce, ready-made sauces, pesto . . . )
34. Eggs
35. Fresh vegetables
36. Pizza
37. Stuffed sandwiches
38. Frankfurters
39. French fries
40. Frozen or deep-frozen products
41. Prepared and precooked meals
42. None of these
43. I don't know

Q6—Think about the last SEVEN days, in your household, how much of the products you indicated did you waste?

| | Less than 100 g/Less Than One Fruit/Less Than One Glass | 100–200 g/Less Than One Fruit And a Half/Less Than One Glass and a Half | 200–300 g/Less Than Two Fruits/Less Than Two Glasses | More than 300 g/More Than Two Fruits/More Than Two Glasses | Nothing/We Don't Consume It | I don't Know |
|---|---|---|---|---|---|---|
| 1. Cold cuts/salami/cured meat | ○ | ○ | ○ | ○ | ○ | ○ |
| 2. Soft drinks (fruit juices, Coke, mineral water, etc.) | ○ | ○ | ○ | ○ | ○ | ○ |
| 3. Alcoholic beverages (wine, beer, etc.) | ○ | ○ | ○ | ○ | ○ | ○ |
| 4. Butter, margarine and oil | ○ | ○ | ○ | ○ | ○ | ○ |
| 5. Cooked WHITE meat (excluding cured meat) | ○ | ○ | ○ | ○ | ○ | ○ |
| 6. Raw WHITE meat (excluding cured meat) | ○ | ○ | ○ | ○ | ○ | ○ |
| 7. Cooked RED meat (excluding cured meat) | ○ | ○ | ○ | ○ | ○ | ○ |
| 8. Raw RED meat (excluding cured meat) | ○ | ○ | ○ | ○ | ○ | ○ |
| 9. Prepared or precooked food (portions, roast chickens, pizzas, etc.) | ○ | ○ | ○ | ○ | ○ | ○ |
| 10. Onions, garlic and tubers (potatoes, carrots, etc.) | ○ | ○ | ○ | ○ | ○ | ○ |
| 11. Sweets (cakes, ice-cream, etc.) | ○ | ○ | ○ | ○ | ○ | ○ |
| 12. Chocolate, spreads, etc. | ○ | ○ | ○ | ○ | ○ | ○ |
| 13. Cheese | ○ | ○ | ○ | ○ | ○ | ○ |
| 14. Fresh fruit | ○ | ○ | ○ | ○ | ○ | ○ |
| 15. Non-fresh fruit and vegetables (jarred, canned) | ○ | ○ | ○ | ○ | ○ | ○ |
| 16. Salads | ○ | ○ | ○ | ○ | ○ | ○ |
| 17. Milk and yogurt | ○ | ○ | ○ | ○ | ○ | ○ |
| 18. Dairy products (mozzarella, cottage cheese … ) | ○ | ○ | ○ | ○ | ○ | ○ |
| 19. Legumes (lentils, beans. chickpeas, etc.) | ○ | ○ | ○ | ○ | ○ | ○ |
| 20. Jams and marmalades | ○ | ○ | ○ | ○ | ○ | ○ |
| 21. Mayonnaise and egg-based sauces (es. Tartar, Bernese … ) | ○ | ○ | ○ | ○ | ○ | ○ |
| 22. Fresh bread | ○ | ○ | ○ | ○ | ○ | ○ |
| 23. Packaged bread | ○ | ○ | ○ | ○ | ○ | ○ |
| 24. Pasta and fresh pasta (raw) | ○ | ○ | ○ | ○ | ○ | ○ |
| 25. Cooked pasta | ○ | ○ | ○ | ○ | ○ | ○ |
| 26. Raw fish/crustaceans/shellfish | ○ | ○ | ○ | ○ | ○ | ○ |
| 27. Cooked fish/crustaceans/shellfish | ○ | ○ | ○ | ○ | ○ | ○ |
| 28. Breakfast products (cookies, cereals, rusks, etc.) | ○ | ○ | ○ | ○ | ○ | ○ |
| 29. Frozen products (veggie soups, etc.) | ○ | ○ | ○ | ○ | ○ | ○ |

| | Less than 100 g/Less Than One Fruit/Less Than One Glass | 100–200 g/Less Than One Fruit And a Half/Less Than One Glass and a Half | 200–300 g/Less Than Two Fruits/Less Than Two Glasses | More than 300 g/More Than Two Fruits/More Than Two Glasses | Nothing/We Don't Consume It | I don't Know |
|---|---|---|---|---|---|---|
| 30. Rice and other cooked grains | ○ | ○ | ○ | ○ | ○ | ○ |
| 31. Rice and other uncooked grains | ○ | ○ | ○ | ○ | ○ | ○ |
| 32. Sauces (e.g., Ketchup, tabasco . . . ) | ○ | ○ | ○ | ○ | ○ | ○ |
| 33. Sauces (es. tomato sauce, ready-made sauces, pesto . . . ) | ○ | ○ | ○ | ○ | ○ | ○ |
| 34. Eggs | ○ | ○ | ○ | ○ | ○ | ○ |
| 35. Fresh vegetables | ○ | ○ | ○ | ○ | ○ | ○ |
| 36. Pizza | ○ | ○ | ○ | ○ | ○ | ○ |
| 37. Stuffed sandwiches | ○ | ○ | ○ | ○ | ○ | ○ |
| 38. Frankfurters | ○ | ○ | ○ | ○ | ○ | ○ |
| 39. French fries | ○ | ○ | ○ | ○ | ○ | ○ |
| 40. Frozen or deep-frozen products | ○ | ○ | ○ | ○ | ○ | ○ |
| 41. Prepared and precooked meals | ○ | ○ | ○ | ○ | ○ | ○ |

Q7—You throw away food mainly because . . .

| | Always | Often | Sometimes | Rarely | Never |
|---|---|---|---|---|---|
| 1. I buy too much | ○ | ○ | ○ | ○ | ○ |
| 2. Too much time passes between groceries and food deteriorates | ○ | ○ | ○ | ○ | ○ |
| 3. I miscalculate the amount of food needed | ○ | ○ | ○ | ○ | ○ |
| 4. I'm always afraid of not having enough food at home | ○ | ○ | ○ | ○ | ○ |
| 5. Fruits and vegetables are often stored in refrigerators and when I bring them home they go bad | ○ | ○ | ○ | ○ | ○ |
| 6. Sold food is already old | ○ | ○ | ○ | ○ | ○ |
| 7. I don't know to preserve and store food | ○ | ○ | ○ | ○ | ○ |
| 8. I buy too big portions and packages | ○ | ○ | ○ | ○ | ○ |
| 9. I cook too much | ○ | ○ | ○ | ○ | ○ |
| 10. I buy food that I don't not like | ○ | ○ | ○ | ○ | ○ |
| 11. There are too many discounts on food products | ○ | ○ | ○ | ○ | ○ |
| 12. I forget about it and it expires/molds/rots/ the smell or taste deteriorates | ○ | ○ | ○ | ○ | ○ |
| 13. I don't like leftovers | ○ | ○ | ○ | ○ | ○ |

Q8—How much do you agree with each of the following statements about food waste?

| | I strongly agree | I agree | I slightly agree | Neither agree nor disagree | I slightly disagree | I disagree | I strongly disagree |
|---|---|---|---|---|---|---|---|
| 1. It is an ethical issue (it is immoral) | ○ | ○ | ○ | ○ | ○ | ○ | ○ |
| 2. It has economic and social consequences | ○ | ○ | ○ | ○ | ○ | ○ | ○ |
| 3. It has environmental consequences | ○ | ○ | ○ | ○ | ○ | ○ | ○ |
| 4. It contributes to global warming | ○ | ○ | ○ | ○ | ○ | ○ | ○ |
| 5. It causes a waste of money for families | ○ | ○ | ○ | ○ | ○ | ○ | ○ |
| 6. It has negative economic consequences on my family | ○ | ○ | ○ | ○ | ○ | ○ | ○ |
| 7. It reduces the production system efficiency | ○ | ○ | ○ | ○ | ○ | ○ | ○ |
| 8. It causes an increase in pollution due to excess waste disposal | ○ | ○ | ○ | ○ | ○ | ○ | ○ |
| 9. It causes an increase in inequalities between rich and poor countries | ○ | ○ | ○ | ○ | ○ | ○ | ○ |
| 10. It has a negative educational impact on young people | ○ | ○ | ○ | ○ | ○ | ○ | ○ |
| 11. It causes an increase in food prices | ○ | ○ | ○ | ○ | ○ | ○ | ○ |
| 12. It causes failures in food distribution affecting those who cannot afford it | ○ | ○ | ○ | ○ | ○ | ○ | ○ |
| 13. It generates waste of valuable resources such as water, energy, and soil | ○ | ○ | ○ | ○ | ○ | ○ | ○ |

Q9—To reduce food waste, those who take care of the household and grocery shopping can adopt several behaviors. Please indicate how often you . . . :

| | Always | Often | Rarely | Never | I don't know |
|---|---|---|---|---|---|
| 1. Make a shopping list | ○ | ○ | ○ | ○ | ○ |
| 2. Plan of what to cook on each day of the week | ○ | ○ | ○ | ○ | ○ |
| 3. Make sure to eat first the food that is about to expire | ○ | ○ | ○ | ○ | ○ |
| 4. Deep-freeze the food that is not consumed right away | ○ | ○ | ○ | ○ | ○ |
| 5. know exactly what is in storage, in the fridge and in the freezer | ○ | ○ | ○ | ○ | ○ |
| 6. Keep the storage, the fridge and the freezer well organized | ○ | ○ | ○ | ○ | ○ |
| 7. Weight ingredients during meal preparation | ○ | ○ | ○ | ○ | ○ |
| 8. Before cooking, carefully evaluate the needed food quantities | ○ | ○ | ○ | ○ | ○ |
| 9. Preserve leftovers from preparations | ○ | ○ | ○ | ○ | ○ |
| 10. Preserve leftovers from preparations | ○ | ○ | ○ | ○ | ○ |
| 11. Any prepared food is sure to be eaten, including leftovers | ○ | ○ | ○ | ○ | ○ |
| 12. If food is just expired (one day) (one day), I check that it is good, and if it is good I use it | ○ | ○ | ○ | ○ | ○ |
| 13. When I eat out (restaurant, canteen) I take home what I can't eat | ○ | ○ | ○ | ○ | ○ |

Q10—In order to reduce your household's food waste, do you adopt any of the following purchasing strategies?

- App/websites for purchasing unsold products to limit food waste
- App/websites for purchasing unsold fresh fruit and vegetables for being damaged/unaesthetic
- App/websites for trading expiring food with neighbors
- App/websites offering recipes by entering the list of expiring products available in the house
- App that track expiration dates and help preparing shopping lists
- Shopping list based on weekly menu
- Purchase of small portions
- Frequent grocery shopping (day-by-day)
- Purchase large quantities of fish, meat and vegetables, dividing them into small/single portions to be frozen
- Prefer long-life products
- Smart refrigerator or shelfs that monitors expiring products
- None of these

Q11—How much useful do you think each of the following measures is in reducing food waste?

| | Very Much Useful | Quite Useful | Mildly Useful | Not at All Useful | I Don't Know |
|---|---|---|---|---|---|
| 1. Make citizen aware of negative environmental impacts | ○ | ○ | ○ | ○ | ○ |
| 2. Make citizen aware of negative economic impacts | ○ | ○ | ○ | ○ | ○ |
| 3. Rise taxation based on food waste | ○ | ○ | ○ | ○ | ○ |
| 4. Charging more for food | ○ | ○ | ○ | ○ | ○ |
| 5. Focus on education in schools | ○ | ○ | ○ | ○ | ○ |
| 6. Make smaller portions | ○ | ○ | ○ | ○ | ○ |
| 7. Make bigger portions | ○ | ○ | ○ | ○ | ○ |
| 8. Improve food labeling | ○ | ○ | ○ | ○ | ○ |

Q12—In your opinion, [COUNTRY] families throw away food because . . . (max 4 answers)

- They buy too much
- Too much time passes between groceries and food deteriorates
- They miscalculate the amount of food needed
- They are afraid of not having enough food at home
- Fruits and vegetables are often stored in refrigerators and when they bring them home, they go bad
- Sold food is already old
- They don't know to preserve and store food
- They buy too big portions and packages
- They cook too much
- They buy food that they don't not like
- There are too many discounts on food products
- They forget about it and it expires/molds/rots/ the smell or taste deteriorates
- They don't like leftovers
- I don't know

Q13—When grocery shopping, do you prefer to buy:

1. Family size packages to save money, even risking waste
2. Small portions to avoid food waste, even if packaging increases
3. Portions with daily consumption amounts clearly stated and recyclable packaging to reduce waste and environmental impacts, even

Q14—1Q What information do you think should be included in food nutrition labels?

- Information on quality of single ingredients
- Information on origins of ingredients
- Information on environmental impacts of products
- Information on nutritional characteristics of products
- Precise information on each ingredient nutritional intake
- Information on in ingredients that may cause allergies
- Information on nutritional intakes that consider the average diet of individuals
- None of these

Q15—To what extent do you think a better labeling mechanism with respect to nutritional values could influence consumer purchases?

- Strongly agree
- Agree
- Disagree
- Strongly disagree
- Don't know

What is you highest education title?

- No title
- Elementary school graduation
- Junior high school graduation
- High school diploma
- Bachelor's degree
- Master's degree
- Postgraduate title

What is your current employment status?

- Full-time worker
- Part-time worker
- Freelancer/self-employed
- Unemployed/Looking for work
- Unemployed and not looking for work/Unable to work
- Housekeeper
- Retired
- Student

Your household income allows you to live . . .

- Very comfortably
- Comfortably
- With some difficulties
- With strong difficulties
- I feel poor and never make ends meet
- I prefer not to answer

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
