# Peer review of "Tell Me What You Waste and I’ll Tell You Who You Are: An Eight-Country Comparison of Consumers’ Food Waste Habits"

_sustainability, doi:10.3390/su15010430_

Round 1

Reviewer 1 Report

The paper is intersting presenting results about the topic of food waste, a domain where data is difficult to collect. For this reason the paper has an important contribution, althogh the results are based on self-reported data.

In order to increase the scientifc soundeness, I have the following suggestions and modifications: 

Describe more clearly the regressions presented in chapter 3.2. It is not clear what are the regressions you have used for the results in the annex for both table 1 and table 2. Describe more in detail, for each part of table 1, which are the dependent and independent variables. I also don't undestant, why you have the value 0 for some variables (Italy, male....). Practically, you should apply the regression in the same way, you did for the other countries. 

Reviewer 2 Report

The manuscript is interesting and deals with a situation that has persisted for a very long time among consumers. There is a need to identify the reasons why a lot of food is lost and to establish the directions of action.

Reviewer 3 Report

This research explores the relationship between household food waste and dietary habits, starting from an original survey conducted in eight countries in 2021. 8,000 questionnaires were recorded from samples representative of adult population of each country through an online survey. Relationship between per capita self-reported amount of food waste (expressed in kilocalories) and self-declared dietary habits (Traditional, Healthy and Sustainable, Vegetarian, Smart, Confused) was estimated using multiple linear regression models. The results suggest that the share of population adopting a smart diet is on average 2.7% of the sample, suggesting that more work needs to be done to reach this goal.

Besides, other minor issues, my main concerns are: 

1)     Literature is not reviewed at all. It requires clarification and justification as to how the authors discovered those gaps. I'm asking how you identified the gaps in your argument and came up with the solutions you did. Please incorporate analysis of at least twenty recent works/studies that are published in reputed journals.

2)     There are so many linguistic and formatting problems with it. A thorough review is advised.

3)     The questionnaire in Appendix A is not viewable. There are formatting issues in it. Please see it.

4)     The questionnaire needs to be discussed in the Introduction section with a subheading. How did you develop the questionnaire? The right and justifiable answer to this question may be provided in that section.  

5)     Please avoid footnotes.

6)     Please assign a meaningful heading to subsection 2.1.

7)     A list of abbreviations may be provided.

8)     Two or three more keywords may be provided.

Round 2

Reviewer 3 Report

Thanks. I'm satisfied.